# A structural and functional subdivision in central orbitofrontal cortex

Maya Zhe Wang 📧 [1,2✉], Benjamin Y. Hayden 📧 [1,2,3] & Sarah R. Heilbronner[1,3]

Economic choice requires many cognitive subprocesses, including stimulus detection, valuation, motor output, and outcome monitoring; many of these subprocesses are associated with the central orbitofrontal cortex (cOFC). Prior work has largely assumed that the cOFC is a single region with a single function. Here, we challenge that unified view with convergent anatomical and physiological results from rhesus macaques. Anatomically, we show that the cOFC can be subdivided according to its much stronger (medial) or weaker (lateral) bidirectional anatomical connectivity with the posterior cingulate cortex (PCC). We call these subregions cOFC*m* and cOFC*l*, respectively. These two subregions have notable functional differences. Specifically, cOFC*m* shows enhanced functional connectivity with PCC, as indicated by both spike-field coherence and mutual information. The cOFC*m*-PCC circuit, but not the cOFC*l*-PCC circuit, shows signatures of relaying choice signals from a non-spatial comparison framework to a spatially framed organization and shows a putative bidirectional mutually excitatory pattern.

---

[1] Department of Neuroscience, University of Minnesota, Minneapolis, MN 55455, USA. [2] Center for Magnetic Resonance Research, University of Minnesota, Minneapolis, MN 55455, USA. [3] These authors contributed equally: Benjamin Y. Hayden, Sarah R. Heilbronner. ✉email: mayawangz@gmail.com

Choosing among rewarding options requires coordination of multiple brain functions spanning sensory perception to valuation and motor output. Among brain regions associated with economic choice, the orbitofrontal cortex (OFC) has attracted the lion's share of attention[1–10]. There is increasing evidence of functional subdivsions within OFC relevant to economic choice. For example, the central OFC (cOFC) is associated with evaluation, value comparison, cognitive mapping, and prospection[6,11–14]. The medial OFC may be more associated with abstract valuation and learning processes[12,15]. The lateral OFC may signal resource availability[16]. These distinctions, based on coarse parcellations, likely reflect just some of the functional heterogeneity present within the OFC.

Economic choice requires the transformation of sensory and mnemonic information into actions[11,17–23]. In other words, economic choice involves a transformation from a non-spatial comparison framework to action-oriented, and therefore in most cases spatial, one. It is likely that the cOFC plays a key role in this process. However, the nature of that role remains unclear. Many studies have emphasized the abstract side of cOFC processing; however, a growing number of studies suggest that it may have an important spatial role as well (e.g.,[24–27]). The inconsistency across studies, along with the functional divisions explained above, raise the possibility that different parts of cOFC may have heterogeneous functions. Delineating that heterogeneity may allow for more precise specification of cOFC's contributions to economic choice.

We hypothesized that the key to understanding the role of cOFC in the transformations associated with choice is through its connectivity with another region involved in economic choice: the posterior cingulate cortex (PCC). This region, located in the posteromedial cortex, has not received the same amount of scholarly scrutiny from decision neuroscientists as cOFC. Nevertheless, the PCC has a confirmed spatial repertoire[28–32] and plays a fundamental economic role[29,33–37]. That is, while PCC has consistent responses to outcomes, those responses are spatially selective, perhaps due to the strong interactions between this region and the parietal cortex and medial temporal lobes[38–40]. Finally, PCC has direct bidirectional communication with OFC[38,41–44]. Because of its potential to help translate choice-related information from cOFC into a spatial domain, we wanted to probe how the cOFC-PCC circuit might facilitate transformations associated with choice.

## Results

**OFC-PCC anatomical connectivity.** To identify the anatomical connectivity between PCC and OFC, we injected the tract-tracer fluororuby in the PCC gyrus, centered at the border between areas 23a and 30 (with some involvement of area 29,[45]). This injection resulted in widespread retrograde and anterograde labeling throughout the anterior and posterior cingulate cortices, parietal lobe (precuneus and intraparietal sulcus), medial temporal lobe (hippocampal formation), and frontal cortex (primarily dorsolateral prefrontal and orbitofrontal cortices). Projections to the OFC were particularly interesting for their specificity: cells and terminal fields were clustered around the medial orbital sulcus (mainly area 13a, but also including lateral 14 O and caudal 11, based on Paxinos et al., 2009;[45] Fig. 1A–D). There were projections to other OFC subregions, but these were noticeably and qualitatively less dense. These results are consistent with other, similarly placed cases from the literature[38,41–44], including several indicating a potentially homologous connection in rats[46]. A second injection targeted the PCC sulcus, and also resulted in labeling around the medial orbital sulcus, although it was less specific (Supplementary Fig. 1).

We concluded that although the PCC does connect with other cOFC subareas (cOFCl), its relationship with the subareas surrounding the medial orbital sulcus (from here on referred to as cOFCm) is unique. We next examined the functional properties of this circuit.

**Behavior and electrophysiology.** We recorded neural activity in all three regions—PCC (recording sites were 14–23 mm posterior to bregma; 0–4 mm lateral to midline, 10–20 mm ventral to surface of the brain), cOFCm (9–12 mm anterior to bregma, 7–11 mm lateral to midline, in the medial orbital sulcus, on the orbital base of the cortex), and cOFCl (9–12 mm anterior to bregma, 12–16 mm lateral to midline, in the orbital gyrus and the medial bank of the lateral orbital sulcus, on the orbital base of the cortex) (Fig. 1C)—while rhesus macaques (*Macaca mulatta*, Subjects P and S) performed an economic choice task we have used several times in the past (first used in ref. [47]; Fig. 2A). On each trial, the subject chose between two randomly generated risky offers (i.e., gambles). Offer varied along the dimension of stakes (small, 125 µL, medium, 165 µL, or large, 240 µL, represented as non-red colors) and probability (0–100% by 1% increments, represented as proportion of the non-red color). Features of each offer and the location (left vs. right) of the first offer (offer 1) were independently randomized on each trial.

As in our past studies using this task (e.g.,[47]), both subjects generally chose the option with higher expected value (EV, Supplementary Fig. 2). Specifically, subjects preferred larger EV on 73.10% of the trials (subject P: 73.35%; subject S: 72.39%, both $p < 0.001$, binomial test). This behavior resembles those we observed in past studies (e.g.,[47,48]). We defined trials in which subjects chose the inferior option as "error trials" (see below). We recorded 44 cells in cOFCm (23 from subject P, 21 from subject S), 54 cells in cOFCl (28 from subject P, 26 from subject S) and 213 cells in PCC (89 from subject P, 124 from subject S). We confirmed recording location by reconciling with atlases based on MRI, and further confirmed by listening to white and gray matter changes when driving down the probe and creating an atlas that was reconciled to the Brainsight images.

To obtain as unbiased as possible a survey of neurons in our population, we performed no preselection on neurons for functional roles. Average firing rates (FR) were low. Firing rate across all times (including inter-trial interval (ITI)) were 1.44 Hz in PCC, 0.70 Hz in cOFCm, and 0.80 Hz in cOFCl. Comparing mean ranks with Kruskal–Wallis test (a.k.a. non-parametric ANOVA), there is a significant difference among three regions ($\chi^2 = 26.46$, $p < 0.001$). PCC had a higher FR than both cOFCm ($p = 0.002$) and cOFCl ($p < 0.001$) while the FR in cOFCm and cOFCl did not differ ($p = 0.814$).

**Functional connectivity.** To ask whether the cOFCm -PCC circuit shows greater functional connectivity than the cOFCl -PCC circuit, we employed spike-field coherence, which relates the recorded action potentials of one region to the local field potential (LFP) oscillations of another. This analysis shows how the spiking region and LFP region communicate and synchronize with each other[49–53]; see Methods and Supplementary Fig. 3). During the offer epoch, the broadband spike-field coherence in the cOFCm_spk-PCC_lfp circuit is higher than that in the cOFCl_spk-PCC_lfp circuit ($z = 5.01$, $p < 0.001$, Wilcoxon signed rank test, Fig. 2B–C). Indeed, cOFCm_spk-PCC_lfp shows higher coherence within all five frequency bands that we tested during the offer epoch (delta = 0.5–5 Hz; theta = 5–10 Hz; alpha = 10–15 Hz; beta = 15–30 Hz; gamma = 30–100 Hz; Fig. 2D; Supplementary material; Methods). The same pattern occurs in the choice and outcome epochs (cOFCm_spk-PCC_lfp > cOFCl_spk-PCC_lfp; choice:

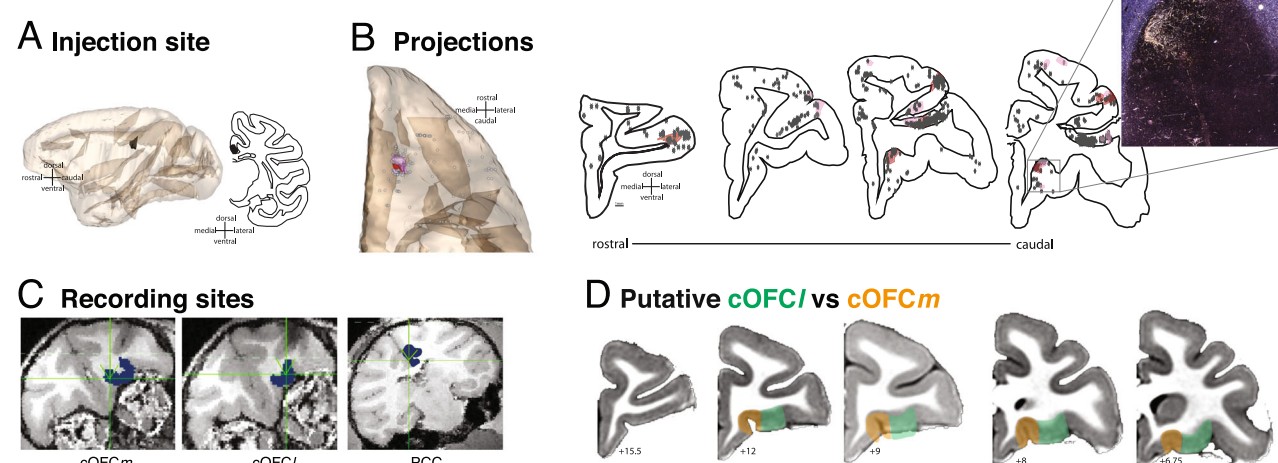

**Fig. 1 Anatomical connectivity between cOFC and PCC and matching recording sites. A** Injection site (black, Case M1FR) is rendered in 3D and shown in a sagittal view (left) and on a coronal slice (right). **B** Projections to the cOFC rendered in 3D and shown on an orbital view. Red indicates dense terminal fields; pink indicates light terminal fields; gray spheres are labeled cells. The majority of cOFC labeling is around the medial orbital sulcus. On the right, coronal slices with full PFC labeling, colors are as on the left. A photomicrograph indicates label around the medial orbital sulcus. **C** Coronal sections of example recording site from each of OFC, with cOFC*m* on left and cOFC*l* in the middle, and PCC. **D** Putative subdivisions of cOFC on coronal slices of an atlas brain[76].

$z = 2.81$, $p = 0.005$; outcome: $z = 3.70$, $p = 0.005$). During choice, higher coherence occurs in the cOFC$m_{spk}$-PCC$_{lfp}$ circuit than the cOFC$l$ $_{spk}$-PCC$_{lfp}$ circuit within the theta, alpha, and gamma bands, but not the delta or beta bands (Fig. 2E; Supplementary material). During outcome, higher coherence occurs in all but the beta band in the cOFC$m_{spk}$-PCC$_{lfp}$ circuit than the cOFC$m_{spk}$-PCC$_{lfsp}$ circuit (Fig. 2F; Supplementary material).

We next examined mutual information between these areas (see Methods), which captures the shared information, in entropy, between two spike trains, one from each region, in either the cOFC$m$-PCC circuit or cOFC$l$-PCC circuit. We first defined one information channel as one spike train from one region (either cOFC$m$ or cOFC$l$) and another spike train from the other region (PCC; using a method developed by ref. [54]). We identified 9372 channels in the cOFC$m$-PCC circuit and 11502 channels in the cOFC$l$ -PCC circuit and calculated the averaged mutual information per channel within each circuit. We found that the cOFC$m$-PCC circuit shares higher mutual information than cOFC$l$-PCC ($7.44 \times 10^{-4}$ vs. $6.72 \times 10^{-4}$ bits/channel; $z = 17.47$, $p < 0.001$, Wilcoxon signed rank test). Mutual information in both circuits increased significantly at task onset ($p < 0.025$, shuffle test; Supplementary material; Methods), suggesting that the observed mutual information effect reflects task-driven, rather than spontaneous, fluctuations (Fig. 2G).

**Neural computation**. We next analyzed encoding of task variables with a multiple linear regression model. All three regions encoded offer and outcome values in their respective epochs with similar proportions of neurons, encoding strengths, and latency (Supplementary material; Methods). They also all encoded the chosen option (offer 1 vs. 2) and chosen location (left vs. right). Note that in this case, latency for chosen offer is defined relative to the moment of the appearance of the second offer. However, cOFC$m$ encoded the chosen option (offer 1 vs. 2) with shorter latency (90 ms, $F = 3.35$, $p = 0.037$, GLM Gamma distribution; Methods) than both cOFC$l$ (170 ms, $t = -2.14$, $p = 0.033$) and PCC (150 ms, $t = -2.36$, $p = 0.019$), suggesting chosen option information arises first in cOFC$m$. PCC appears to be more spatially sensitive than either OFC region: it showed a higher proportion of neurons encoding chosen location (19.25%, $n = 41/213$, $p < 0.001$, binomial test) than chosen

option (10.80%, $n = 23/213$, $p = 0.001$, binomial test; these are different, $\chi^2 = 5.31$, $p = 0.021$, chi-square test); neither OFC region shows this pattern (cOFC$m$: 18.8% encoding location, 18.8% encoding option; cOFC$l$: 13.0% encoding location, 16.7% encoding option, see Supplementary material). In addition, PCC (140 ms) and cOFC$m$ (150 ms) encoded the chosen location with significantly shorter latencies than cOFC$l$ (230 ms; $F = 5.71$, $p = 0.004$; Supplementary material).

Since chosen option encoding arises first in cOFC$m$, this result suggests that cOFC$m$ might carry out the value comparison between two offers to arrive at the choice encoding. To explore this idea more fully, we turned our focus to the negative correlation of regression coefficients for the two offers, which we have argued is a signature of comparison[47,55]. The reason this is a putative signal of value comparison is that it reflects a coding of the difference in the values of the two options—the key decision variable for choice, because it can be rectified to produce choice[19]. We performed this analysis using a 200 ms analysis window (350 ms after offer 2 onset) and found that cOFC$m$ showed this putative mutual inhibition signal ($r = -0.36$, $p = 0.016$, Spearman correlation; Fig. 3A). We did not observe such an effect in cOFC$l$ ($r = -0.18$, $p = 0.190$; Fig. 3B) or in PCC ($r = 0.02$, $p = 0.943$; Fig. 3C). We also did not find this negative correlation during the later choice epoch (from 400 ms to 200 ms before choice action) in any of the three regions (Supplementary Fig. 4A–C). The effect size of these negative correlations was not significantly different in cOFC$m$ vs. cOFC$l$ ($z = -0.93$, $p = 0.176$; Fisher's Transformation test) but was significantly larger in cOFC$m$ than in PCC ($z = -2.32$, $p = 0.010$).

This negative correlation between regression weights, then, is a putative neural correlate of value comparison through mutual inhibition. We wondered whether the transition of attention from offer 1 to offer 2 results in a reversal of tuning for offer 1 value, as predicted by attentional alignment models of value encoding[19,56–58]. We found that in cOFC$m$, the relevant betas are positive correlated ($r = 0.385$, $p = 0.010$). Likewise, the regression weights for offer 1 during the offer 1 epoch were also negatively correlated with those for offer 2 during the offer 2 epoch ($r = -0.314$, $p = 0.039$). Consistent with the idea that the two cOFC subregions are functionally different, the pattern was different in cOFC$l$—specifically, no correlation was observed for

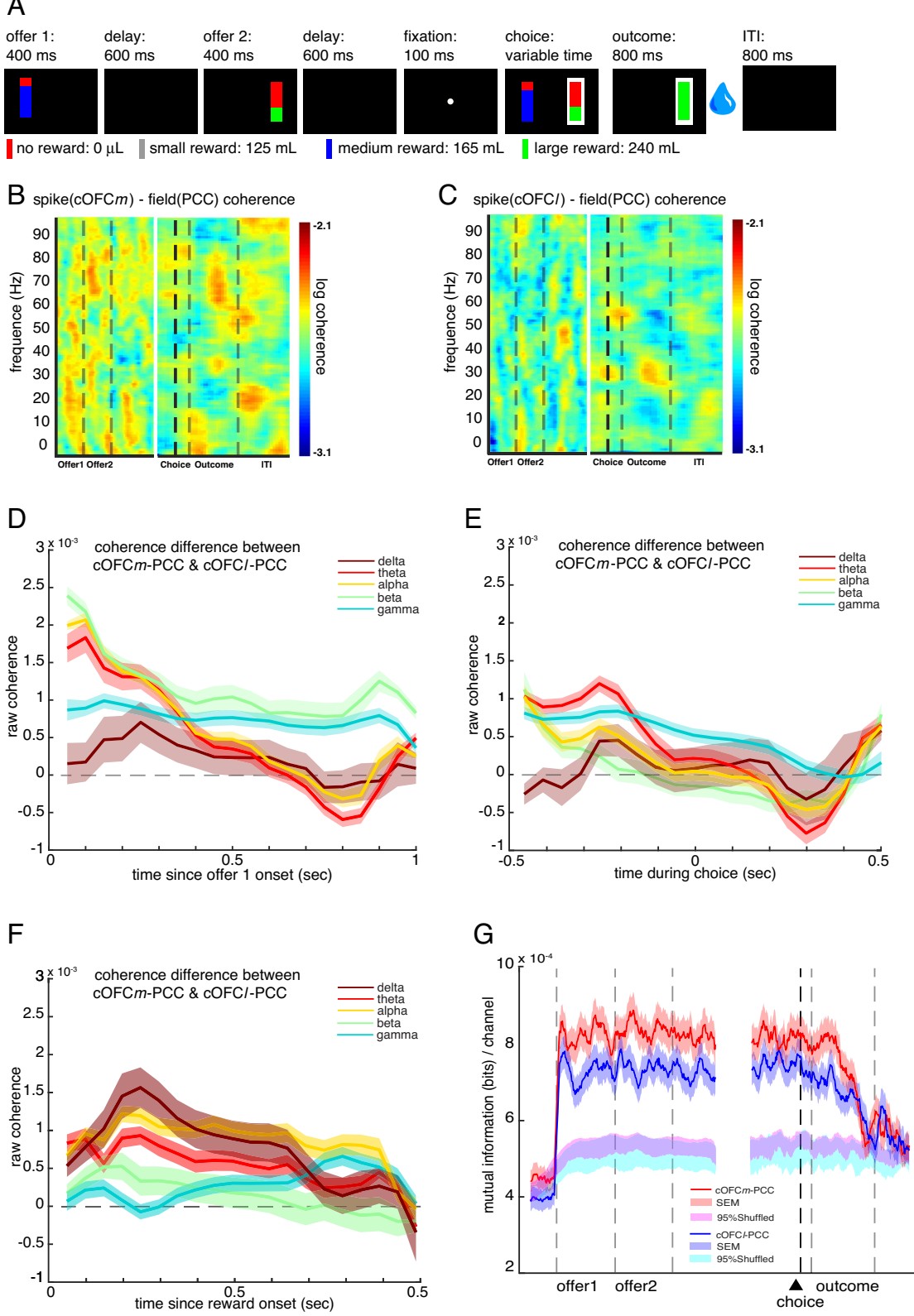

either comparison (respectively: $r = 0.078$, $p = 0.578$) and $r = 0.224$, $p = 0.104$). The corresponding data in PCC resembled the patterns in cOFCm for the first comparison, although not for the second ($r = 0.271$, $p < 0.001$; $r = 0.065$, $p = 0.344$). Overall, these results highlight the differences between cOFCl and cOFCm, specifically, that the putative neural correlate of value comparison is observed in the medial area, but not detected in the lateral area.

**Comparison in a spatial frame of reference in PCC.** We observed a comparison signal in PCC, but as befitting its supposed spatial role, it was a spatially oriented value comparison. Specifically, we observed a negative correlation between regression coefficients for *left* and *right* offer values (EV*l* and EV*r*), as opposed to first and second as in the previous analysis ($r = -0.24$, $p < 0.001$; Fig. 3F) during the offer 2 epoch in PCC. This signal

**Fig. 2 Task and functional connectivity. A** Two-option risky choice task. Black rectangles symbolize various task epochs subjects experience during task. Stakes are represented as different colors: small (gray), medium (blue), or large (green) reward. Losing the gamble (no reward) is represented in red. The height of the stakes-color region represents the probability of winning the gamble, and the height of the red-color region represents the probability of losing the gamble. The white frame around the right option in the choice epoch represents the scenario where the subject chooses the right option with eye-fixation. The water droplet symbol indicates that reward delivery (or lack thereof) occurs. **B** Trial-averaged spike-field coherence in cOFC$m$ $_{spk}$-PCC$_{lfp}$ circuit. *X*-axis: time in a trial. *Y*-axis: frequency. Color: strength of spike-field coherence on log10 scale (warmer colors = higher coherence). Data from the first half of the trial (offer period) was aligned at offer 1 onset. Data from the second half of the trial (choice period) was aligned at choice execution. Dotted gray vertical lines mark the boundaries of epochs. Solid black vertical line marks the moment a choice was made. **C** Spike-field coherence in cOFC$l$ $_{spk}$-PCC$_{lfp}$. Conventions as in (**B**). **D–F** Difference in spike-field coherence between the two circuits (coherence in cOFC$m$ $_{spk}$-PCC$_{lfp}$ circuit minus coherence in cOFC$l$ $_{spk}$-PCC$_{lfp}$ circuit), broken down into different frequency bands as a function of time (*Methods*), during (**C**) offer 1 epoch, (**D**) choice epoch, and (**E**) reward epoch. **G** Mutual information (averaged across number of channels) in cOFC$m$ -PCC and cOFC$l$ -PCC circuits. SEM: standard error of the mean. Red shaded area: SEM of mutual information in cOFC$m$ -PCC circuit. Blue shaded area: SEM of mutual information in cOFC$l$ -PCC circuit. Magenta and cyan shaded areas: the middle 95% range of the randomly shuffled mutual information (500 times) for cOFC$m$ -PCC and cOFC$l$ -PCC circuits, respectively. Thus, the original (non-shuffled) mutual information values outside of the shaded area is significantly higher/lower than expected by chance.

was not significant in either cOFC$m$ ($r = -0.16$, $p = 0.293$; Fig. 3D) or cOFC$l$ ($r = 0.10$, $p = 0.454$, Fig. 3E). This result suggests that even though both cOFC$m$ and PCC are involved in value comparison, they adopted different frameworks: cOFC$m$ computed the choice in a more abstract space consistent with the order in which options appeared in sequence, but PCC computed the same choice in a more concrete action space. Interestingly, the effect size of these negative correlations was not significantly different in cOFC$m$ vs. PCC ($z = -0.49$, $p = 0.313$) but was significantly larger in PCC than in cOFC$l$ ($z = -2.21$, $p = 0.014$). However, during the later choice epoch, we found the same signal in both PCC ($r = -0.19$, $p = 0.006$) and cOFC$m$ ($r = -0.33$, $p = 0.029$), but not cOFC$l$ ($r = 0.31$, $p = 0.022$) (Supplementary Fig. 4D–F), suggesting that a comparison signal in the spatial framework emerges first in PCC and later in cOFC$m$, but not in cOFC$l$.

**Transformation of comparison signal.** Next, we asked how the observed functional connectivity between cOFC$m$ and PCC relates to the negotiation between abstract sequence space and action space (that is, between EV1 - the expected value of the first offer - vs. EV2 - the expected value of the second offer) and between left vs. right). We next used Granger causality (see Methods), a method that examines the relative correlation between two time series at different lags to identify a putative causal role between the two, given certain assumptions. We used a 200 ms sliding window over the whole period of interest, that is, an epoch beginning with the appearance of the second offer and ending with the occurrence of the choice, as indicated by the start of a saccade toward the choice target. We found that the strength of comparison signal between EV1 and EV2 in cOFC$m$ Granger-caused the strength of comparison signal between EV$l$-EV$r$ in PCC ($gc = 40.56$, $p = 0.019$) with a 240 ms (4.17 Hz) lag. In the reverse direction, the strength of mutual inhibition for EV$l$-EV$r$ in PCC Granger-caused the strength of mutual inhibition for EV1-EV2 in cOFC$m$ ($gc = 59.75$, $p = 0.014$), but with a much longer lag (380 ms; 2.63 Hz).

Further supporting a functional distinction between cOFC$m$ and cOFC$l$, the strength of mutual inhibition for EV1-EV2 in cOFC$l$ did not Granger-cause the strength of mutual inhibition for EV$l$-EV$r$ in PCC with any time lag (see Methods for controls for confounding variables). These results suggest that through the communication in the cOFC$m$ -PCC circuit, but not the cOFC$l$ -PCC circuit, the computation for value comparison transformed from abstract sequence space (in cOFC$m$) to action space (in PCC).

These results suggest we should be able to decode choice signal more strongly in abstract sequence space (in the format of chosen option, offer 1 vs. 2) in cOFC$m$ but decode choice more strongly in action space (in the format of chosen location, left vs. right) in PCC. This prediction is borne out by Linear Discriminant Analysis (LDA) supports. Although chosen options (offer 1 vs. 2) and chosen location (left vs. right) were all significantly decodable from all three regions, PCC showed a significantly higher decodability for chosen location ($\chi^2 = 8.12$, $p = 0.004$; Fig. 3G±H). More importantly, the decodability for chosen option (offer 1 vs. 2) in cOFC$m$ Granger-caused the decodability for chosen location (left vs. right) in PCC ($gc = 11.19$, $p = 0.025$) with a 200 ms (5 Hz) lag. This Granger-causal relation was absent on error trials ($gc = 3.04$, $p = 0.552$; Supplementary Fig. 6).

In the reverse direction, the decodability for chosen location (left vs. right) in PCC Granger-caused the decodability for chosen option (offer 1 vs. 2) in cOFC$m$ ($gc = 17.59$, $p = 0.025$), but with a longer lag (400 ms; 2.5 Hz). In contrast, the decodability for chosen offer (offer 1 vs. 2) in cOFC$l$ did not Granger-cause the decodability for chosen location (left vs. right) in PCC at any time lag (see Methods for controls for confounding variables). As a control analysis, the same result pattern was not observed for decodability of EV1 (high vs. low; Supplementary Fig. 6; Supplementary material). These results suggest that the cOFC$m$-PCC circuit, but not the cOFC$l$-PCC circuit, mediates the transformation of choice readout from an abstract non-spatial to an action-oriented spatial framework. Speculatively, this transformation may be important for correct choice behavior, since the both the decodability for choice and the Granger causal relation between cOFC$m$ and PCC was disrupted in error trials, and the transformation was impacted by whether the choice was easy or difficult (Supplementary material; Supplementary Fig. 6).

**Population dynamics reflecting the translation to action space.** We asked whether the population activity dynamics[59–63] also reflect the translation of choice to action space in the cOFC$m$-PCC circuit. We conducted PCA on trial-averaged population states for each region and then projected the trial-averaged population activity onto the top-N principal component (PC) space that cumulatively explained >70% of the variance (Methods; we developed this approach in ref. [14]). The projected population trajectories reflect the generative temporal evolution of population dynamics (Fig. 4A–F), and the separation between trajectories, which distinguished task parameters, became significantly higher than shuffled chance level (bottom shaded area) as the trial unfolded. These distinctions diminished in error trials, suggesting that the population dynamics and their separation are indeed crucial for generating correct choice behavior (see Supplementary material for further results supporting this possibility).

We then projected the trial-by-trial population states onto this top-N PC space to obtain trial-by-trial population trajectories and

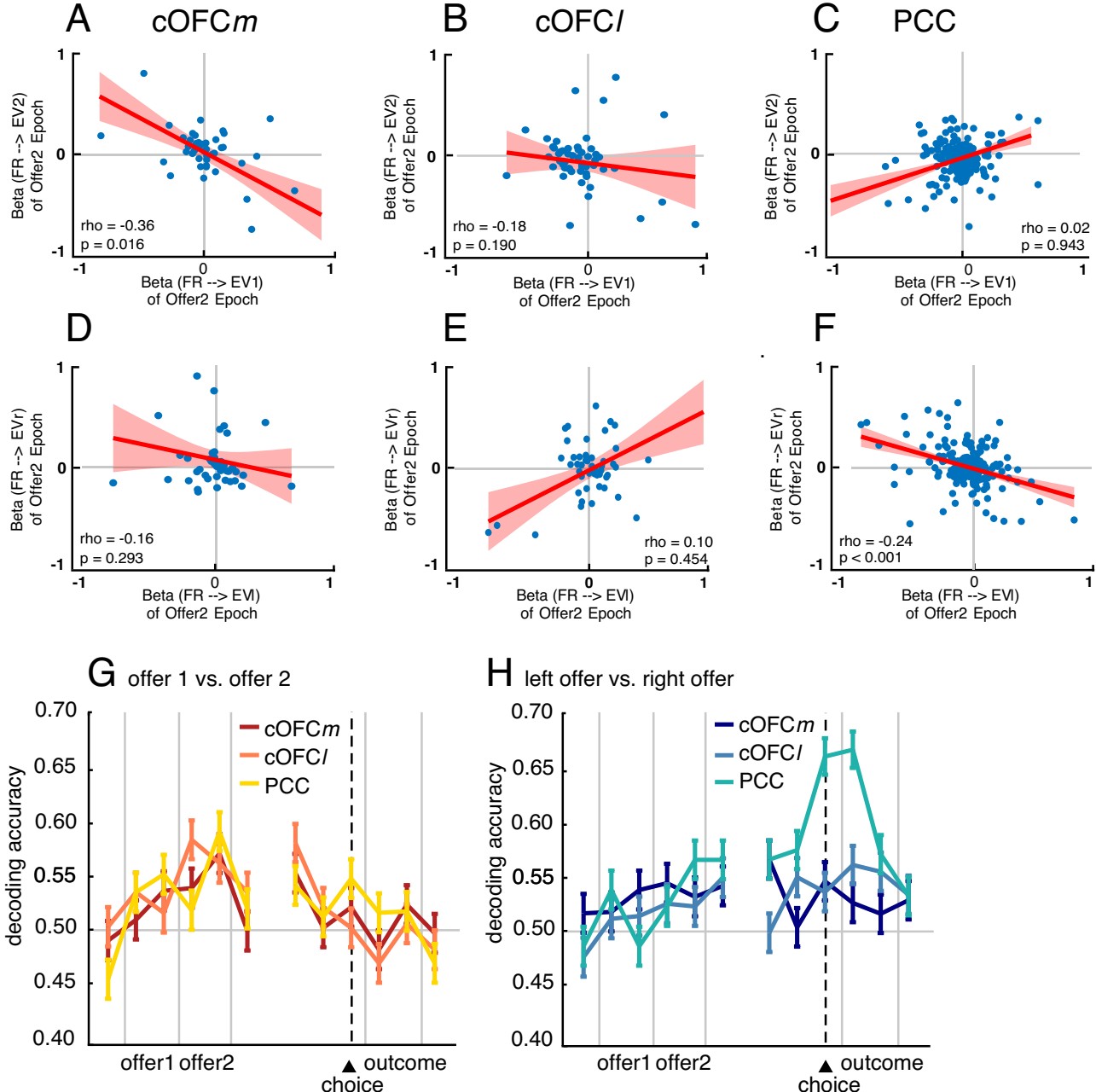

**Fig. 3 Neural computations. A–F** Scatter plots demonstrating population spreads for regression coefficients. Each dot represents one neuron; abscissa and ordinate represent regression coefficients for distinct (and uncorrelated) regressions. Solid red line: fitted Spearman correlation. Shaded area: 95% confidence interval. EV1 = expected value of first offer; EV2 = expected value of second offer. **A–C** Putative mutual inhibition effects (Strait et al., 2014). *Y*-axis indicates regression coefficient for expected value of offer 2 regressed against firing rate in epoch 2. *X*-axis indicates regression coefficient for expected value of offer 1 against firing rate in epoch 2. **D–F** Putative mutual inhibition effects in a spatial framework (new analysis developed for this project): *Y*-axis: regression coefficient for expected value of right offer against firing rate in epoch 2. *X*-axis: regression coefficient for expected value of left offer against firing rate in epoch 2. **A**, **D** cOFC*m*. **B**, **E** cOFC*l*. **C**, **F** PCC. **G**, **H** Decoding accuracy of choice (**G** is accuracy for offer 1 vs offer 2; **H** is accuracy for left vs right offer) based on firing rates using linear discriminant analysis. All simultaneously recorded cells from a single region are used (with no replication) in the decoding analysis. *N* = 805 correct trials. *Y*-axis: probability of decoding correctly. *X*-axis: time in a trial. Error bar: standard error of the mean.

used adjusted distance to measure the trajectory separation (Eq. 1; Methods;[64]). We found significantly larger trajectory separation for chosen option (offer 1 vs. 2) in cOFC*m* ($\chi^2 = 11.51$, $p = 0.003$, Kruskal–Wallis test with Tukey–Kramer multiple comparison) than in cOFC*l* (cOFC*m* > cOFC*l*: $p = 0.007$) and PCC (cOFC*m* > PCC: $p = 0.012$; no significant difference between cOFC*l* and PCC, $p = 0.988$; Fig. 4G). This result highlights the specific role of cOFC*m* in mediating abstract comparison.

In contrast, we found significantly larger trajectory separation for chosen location (left vs right) in PCC ($\chi^2 = 6.27$, $p = 0.043$, Kruskal–Wallis test with Tukey–Kramer multiple comparison) than in cOFC*l* (PCC > cOFC*l*: $p = 0.043$) but not in cOFC*m* (PCC ≈ cOFC*m*: $p = 0.829$; there was no significant difference between cOFC*m* and cOFC*l*, $p = 0.164$; Fig. 4H). There was also no such cross-region distinction for EV1 (high vs. low; Supplementary material). The trajectory separation differences

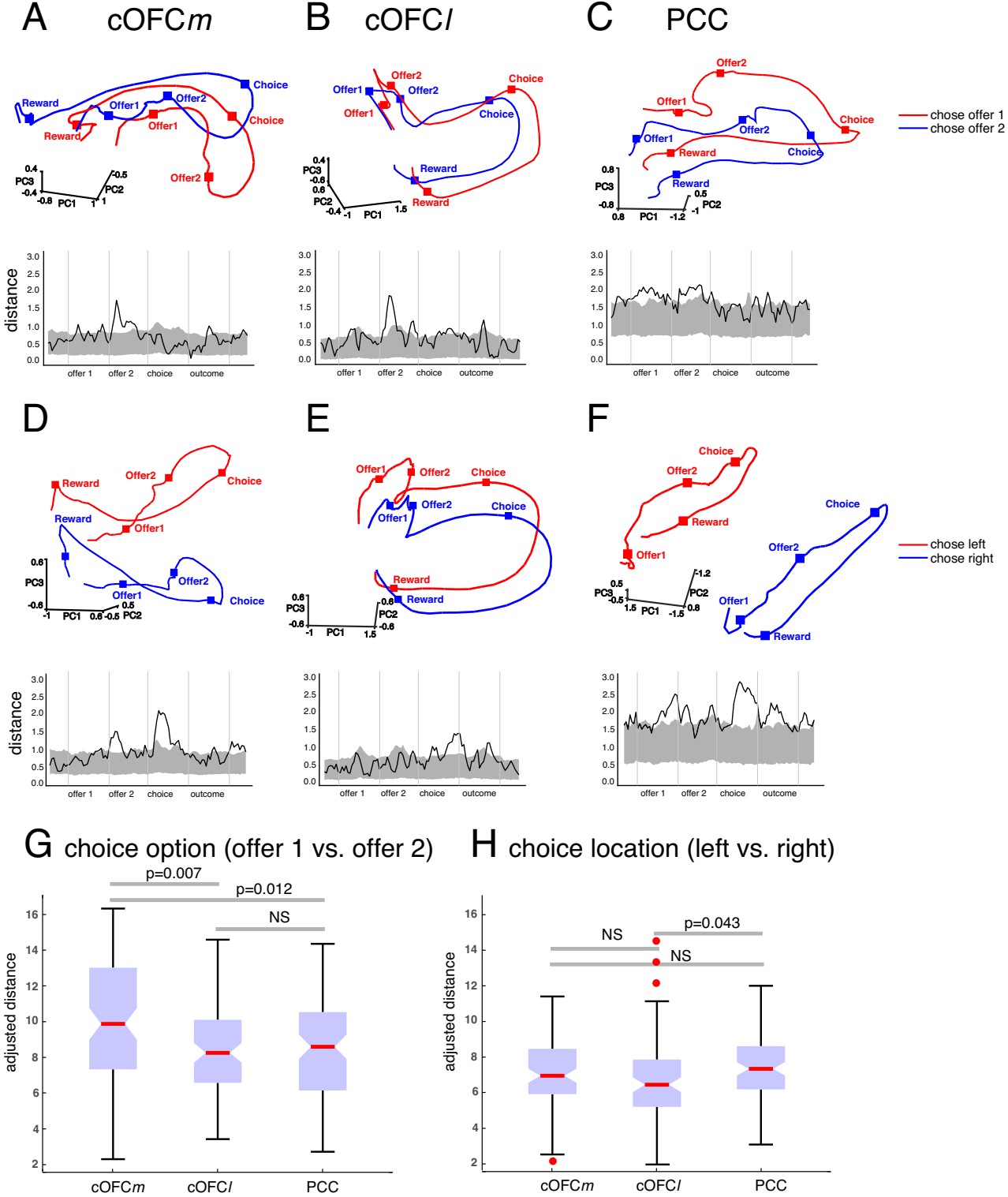

for chosen option and chosen location were also absent in error trials (Supplementary material), consistent with the intuitive idea that the areal difference in the unfolding trajectory separation contributes to correct choice behavior.

Crucially, the separation between population trajectories for chosen option (offer 1 vs. 2) in cOFC*m* Granger-caused the separation between population trajectories for chosen location (left vs. right) in PCC (gc = 9.98, *p* = 0.019), with a 150 ms (6.67 Hz) lag. In the reverse direction, the distance between

population trajectories for chosen location (left vs. right) in PCC Granger-caused the distance between population trajectories for chosen option (offer 1 vs. 2) in cOFC*m* (gc = 17.28, *p* = 0.016) but with a much longer lag (350 ms; 2.86 Hz). Interestingly, this "feedback" influence seems to amplify the cOFC*m* to PCC influence 300 ms after the first instance of Granger causal influence, by increasing the Granger-causality from cOFC*m* to PCC (gc = 38.29, *p* < 0.001; lag = 450 ms; 2.22 Hz). In contrast, the distance between population trajectories for chosen option

**Fig. 4 Different neural dynamics for different subregions.** Top plots: trial averaged population activity projected onto top-N PC space (only top-3 PCs are shown here), separated by choice option (offer 1 vs. 2) (A-C) or choice location (D–F), in cOFCm (left column), cOFCl (middle column), and PCC (right column). Red: trial averaged population activity for choosing offer 1 (**A–C**) or left offer (**D–F**). Blue: trial averaged population activity for choosing offer 2 (**A–C**) or right offer (**D–F**). Squares on the trajectories mark the center of the event epoch window. Bottom plots: separation measured by Euclidean distance between averaged population trajectories (red and blue colored lines). Y-axis: Euclidean distance. X-axis: time in a trial. Dark line: distance between trial-averaged trajectories for choosing offer 1 vs. offer 2 (**A–C**) or choosing left vs. right offer (**D–F**). Shaded area: middle 95% trial-averaged Euclidean distance between population trajectories from condition-shuffled data. Shuffle was only based on the choice of offer 1 or offer 2 (**A–C**) or on the choice of left or right offer (**D–F**), the cell identities and temporal orders were not shuffled. Euclidean distance (i.e., separation; dark line) beyond the shaded area is significant ($p < 0.05$). Specifically, the distance (dark line) larger than (above) the shaded area is where separation between population trajectories is significantly larger than expected by chance ($p < 0.025$). These significant portions mark when the population activity dynamics significantly reflected the choice of offer 1 or offer 2 (**A–C**) or on the choice of left or right offer (**D–F**). **G, H** ranked trial-by-trial adjusted distance. All simultaneously recorded cells from a single region are used (with no replication) to generate trial-by-trial PCA trajectory, with which we measured the adjusted distance. $N = 805$ correct trials. Kruskal–Wallis box plot. The red horizontal line: the median. The bottom and top edges of the box: the 25th and 75th percentiles. The whiskers extend to the most extreme data points not considered outliers. Red dots: individual outliers.

(offer 1 vs. 2) in cOFCl did not Granger-cause the distance between population trajectories for chosen location (left vs. right) in PCC with any time lag (see Methods for the control for confounding variables).

## Discussion

Here we report the existence of two functionally distinct subregions within the cOFC that can be differentiated by their connectivity with the PCC, both anatomically and functionally. cOFCm, located on the banks of the medial orbital sulcus, and cOFCl situated lateral to cOFCm. These two subregions are distinguished anatomically by their connections with PCC. The region we call cOFCm has a stronger anatomical connectivity with PCC than cOFCl. This anatomical distinction corresponds to discrete functional differences, and in particular, suggests a circuit-specific functional separation that relates to the negotiation between cOFCm's abstract (non-spatial, potentially value-based) and PCC's action-based (spatial) modalities. The influence between these two structures is bidirectional, suggesting that both representational frameworks mutually influence each other.

Our data support the hypothesis that within both regions, a mutually inhibitory local circuit exists to compare offers, albeit in different comparison frameworks in cOFCm and in PCC. While local neural computation generates choice representations, their unfolding population dynamics also interact with the generative dynamics in other regions. The Granger-causal relations reported here suggest the possibility that the cOFCm local computation (possibly reflected in the value comparison signal) pushes its own population dynamics to gradually occupy distinct neural subspace for easy readout of choice option (choosing offer 1 vs. offer 2) in the abstract (potentially choice value) space. This cOFCm dynamic then nudges (Granger-causes) the PCC local computation to be conducted along choice action space and pushes its own population dynamics to gradually occupy distinct neural subspace for easy readout of choice action (choosing left vs. right), coinciding with theta oscillations. The PCC dynamic in action space, in turn, strengthens the cOFCm dynamic in abstract space, and the cOFCm dynamic later further amplifies the PCC dynamic in action space, coinciding with delta oscillations. Speculatively, this locally inhibitory (within cOFCm and PCC) and globally excitatory (between cOFCm and PCC) circuit-wise computation pattern, locked with theta and delta band oscillations, potentially translate value representation into choice action by amplifying choice signal through circuit interaction. Moreover, we did not see the information relay between cOFCm and PCC in error trials, suggesting that the transformation of choice in the cOFCm-PCC circuit is essential for generating a correct choice. These circuit interactions (granger causal relation) also differ in easy vs. difficult choices, suggesting a tight relation to choice

behavior. Presumably, after the relay of information between cOFCm and PCC, a downstream area could use the action-bounded choice signal to form an action plan.

Our data suggest that there may be important information traveling from the PCC to the cOFC, and that this information transfer may occur after the transfer of information from cOFC to PCC. It is reasonable, then, to wonder what function this back-transfer serves. Absent causal manipulations, it is impossible to offer a definitive answer. However, our data do provide enough information for us to make an educated speculation. Specifically, we conjecture that the transmission of information from PCC to cOFC can facilitate the process of reaching a consensus within OFC. We have proposed in the past that decisions in core economic regions can occur gradually, and that it is possible to detect partially completed decisions[55,65]. These partially completed decisions could then be transmitted to other regions and, in turn, influence the ongoing decision. While this idea is consistent with our data, our data nonetheless do not offer strong evidence in its favor; as such, testing this hypothesis remains an important future goal.

Traditional approaches to neurophysiology take the classic numbered anatomical areas as homogeneous, and seek to delineate their functions by sampling from them. The anatomical areas are, however, heteromorphic. Specifically, connectivity-based methods, which presumably relate to function more directly than cell-type-based methods, point to important divisions within areas. Thus, while the cOFC is often treated as a single region, our neuroanatomy demonstrates a clear division between the more medial cOFCm and the more lateral cOFCl. These subregions tend to be grouped together in nearly all neurophysiological studies of OFC, including our own past work. However, doing so risks combining different subregions with qualitatively different functions and producing misleading characterizations of regional organization. To speculate, even just within the OFC, we might expect to find subregional organization on the basis of connectivity with the dorsolateral prefrontal cortex (perhaps to assign values to abstract categories), the anterior cingulate cortex (perhaps to use values to update future behavior), and the hippocampus (perhaps to build values from prior associations), among others. These results, then, demonstrate the critical need for greater tract tracing studies, and for further integration of neuroanatomy with single unit electrophysiology in the future.

## Methods

**Neuroanatomy studies**. We injected the bidirectional tracer fluororuby into the PCC of two adult male rhesus macaque (*Macaca mulatta*) subjects. In one (M1FR), the injection site was located at the border of areas 23 and 30 (with some involvement of area 29). In another (M6FR), the injection site was located at the border of areas 23 and 31. We note that, although the PCC is often defined as areas 23 and 31, with areas 29 and 30 instead defined as retrosplenial cortex[66–68], we were

interested in the functionality of this entire caudal cingulate region. Thus, like some prior authors[69–73], here we defined PCC as areas 23, 31, 29, and 30.

Prior to surgery, anatomical T1 and T2-weighted MRIs (3 T for M1FR and 10.5 T for M6FR) were obtained at University of Minnesota's Center for Magnetic Resonance Research. Stereotaxic earbars were filled with Vitamin E solution to visualize on the MRI and guide tracer placement relative to stereotaxic space.

On the day of surgery, monkeys were tranquilized by intramusculuar injections of ketamine (10 mg/kg), midazolam (0.25 mg/kg) and atropine (0.04 mg/kg). A surgical plane of anesthesia was then maintained via the administration of inhalation of isoflurane (1–3%). Monkeys were placed in a stereotaxic instrument (Kopf Instruments), a midline scalp incision was made, and the muscle and fascia were displaced laterally to expose the skull. A craniotomy (~2–3 cm$^2$) was made over the PCC, and small dural incisions were made only at injection sites. Both monkeys received injections of FR (50 nl, 10% in 0.1 M PB, pH 7.4, Invitrogen) in the PCC, as well as injections of additional tracers (lucifer yellow, fluorescein, wheat germ agglutinin conjugated to horseradish peroxidase) in other regions not described here. These do not cross-react with FR and were made distant from the PCC site. Tracers were pressure-injected over 10 min using a 0.5 µl Hamilton syringe. Following each injection, the syringe remained in situ for 20–30 min. Twelve to 14 days after surgery, monkeys were again deeply anesthetized and perfused with 4 L of saline followed by 6 L of a 4% paraformaldehyde/1.5% sucrose solution in 0.1 M PB, pH 7.4. Brains were postfixed overnight and cryoprotected in increasing gradients of sucrose (10, 20, and 30%). Serial sections of 50 µm were cut on a freezing microtome into cryoprotectant solution.

One in eight sections was processed free-floating for immunocytochemistry to visualize the tracer. Tissue was incubated in primary anti-FR (1:6000; Invitrogen) in 10% NGS and 0.3% Triton X-100 (Sigma-Aldrich) in PO4 for 4 nights at 4 °C. After extensive rinsing, the tissue was incubated in biotinylated secondary antibody followed by incubation with the avidin-biotin complex solution (Vectastain ABC kit, Vector Laboratories). Immunoreactivity was visualized using standard DAB procedures. Staining was intensified by incubating the tissue for 5–15 s in a solution of 0.05% DAB tetrahydrochloride, 0.025% cobalt chloride, 0.02% nickel ammonium sulfate, and 0.01% H2O2. Sections were mounted onto gel-coated slides, dehydrated, defatted in xylene, and coverslipped with Permount.

Using a Zeiss M2 AxioImager, light microscopy was used to outline brain sections, PCC injection sites, frontal cortical terminal fields, and frontal cortical labeled cells on 1 in 24 sections (1.2 mm apart). Terminal fields were outlined in darkfield using a 2.0, 4.0, or 10× objective with Neurolucida software (MicroBrightField Bioscience). Terminal fields were considered dense when they could be visualized at a low objective (2.6×)[74]; otherwise, terminal fields were considered sparse. Thin, labeled fibers containing boutons were marked as terminating; thick fibers without boutons were considered passing. Retrogradely labeled cells were identified under brightfield microscopy (20×) using StereoInvestigator software (MicoBrightField Bioscience).

Cases were registered and rendered in 3D in the following way. For each case, a stack of 2D coronal sections was created from its Neurolucida chartings. This stack was imported into IMOD (Boulder Laboratory for 3D Electron Microscopy,[75]), and a 3D reconstruction that contained the injection sites, terminal fields, and cells was created for each case separately. To render these and merge cases together, we used a reference model from the NIMH Macaque Template[76], imported into IMOD. Placement of all contours—injection sites, terminal fields, cells, area outlines—were assessed according to cortical and subcortical landmarks in the brain, then checked with the original case and corrected as needed.

## Neurophysiology studies

*Subjects.* Two male rhesus macaques (*Macaca mulatta*) served as subjects to the neurophysiology experiment. All animal procedures were[77,78] approved by the University Committee on Animal Resources at the University of Rochester (neurophysiology studies) and by the Institutional Animal Care and Use Committee at the University of Minnesota (neurophysiology and neuroanatomy studies). The experiments were designed and conducted in compliance with the Public Health Service's Guide for the Care and Use of Animals. These subjects were used in past studies involving set shifting and risky choice[79–81].

*Behavioral task.* Subjects performed a two-option gambling task identical to the one we used in a previous investigation (Fig. 1, Strait et al., 2014, Yoo and Hayden, 2020; see ref. [82] for context). On each trial, two offers were presented. Each offer was represented by a rectangle on the screen (300 pixels tall and 80 pixels wide); 11.35° of visual angle tall and 4.08° of visual angle wide). Offers were separated from the central fixation point by 550 pixels (27.53° of visual angle). Options were either a gamble or a safe (100% probability) bet for liquid reward. Gamble offers varied in both potential reward size and probability, which were selected with uniform probabilities and independently of one another for each offer and trial. Each gamble rectangle had a red section and a second section that was either blue or green. Blue indicated that the win outcome was a medium size reward (165 µL liquid reward); green indicated that the win outcome was a large reward (240 µL liquid reward). The size of the blue or green portions indicated the probability of winning this medium or large reward (Fig. 1). Win probabilities were drawn from a uniform distribution between 0% and 100%. Reward probabilities were drawn from uniform distributions with resolution only limited by the size of the screen's pixels,

which let us present hundreds of unique gambles. Safe offers (1 out of every 8 offers) were entirely gray. Selecting one would result in a small reward (125 µL) with 100% certainty. Offer reward sizes were selected at random and independent of one another with a 43.75% probability of blue (medium reward) gamble, a 43.75% probability of green (large reward) gambles, and 12.5% probability of safe offers. Note that this means two offers with the same reward size could be (and often were) presented in the same trial.

The sides of the first and second offer (left or right) were randomized on each trial. Each offer appeared for 400 ms followed by a 600 ms empty screen. After the offers were sequentially presented, a central fixation point appeared, and the monkey fixated on it for 100 ms. Then both offers appeared simultaneously and the animal indicated its choice by shifting gaze to its preferred offer and maintaining fixation on it for 200 ms. Failure to maintain fixation would return the monkey to a choice state. Thus, subjects could change their mind if they did so within 200 ms (although they seldom did). Following a successful 200 ms fixation, the chosen offer was immediately outlined with a white frame, the gamble was immediately resolved, and the liquid reward was delivered. When the subject won the gamble, the stake (blue or green) color would fill the offer rectangle while the water aliquot corresponding to the stake color was delivered. When the subject lost the gamble, the red color would fill the offer rectangle while a water reward was omitted (Fig. 2A). Trials that took >7 s were considered aborted due to inattention and were excluded from analysis (this removed <1% of trials). Each trial was followed by an 800 ms ITI with a blank screen.

*Eye tracking and reward delivery.* Eye position was sampled at 1000 Hz by an infrared eye-monitoring camera system (SR Research). Stimuli were controlled by a computer running MATLAB (Mathworks) with Psychtoolbox[83] and Eyelink[84] Toolbox. A standard solenoid valve controlled the duration of fluid reward delivery. For part of the behavioral training, subjects received grape juice or cherry coke instead of water as reward. However, water reward was used during all neural recording sessions. The relationship between solenoid open time and water volume was established and confirmed before, during, and after recording.

*Recording sites.* Two Cilux recording chambers (Crist Instruments) were placed over cOFC and PCC[45,85–87]; Fig. 1D). Note that this posterior region is overlapping with but ventral to a region we have previously recorded in known as CGp[34,88,89]. Position was verified by magnetic resonance imaging with the aid of a Brainsight system (Rogue Research Inc.) for subject P and Cicerone system (Dr. Matthew D. Johnson at University of Minnesota) for subject S. Neuroimaging was performed at the Rochester Center for Brain Imaging, on a Siemens 3 T MAGNETOM Trio Tim using 0.5 mm voxels. We confirmed recording locations by listening for characteristic sounds of white and gray matter during recording, which in all cases matched the loci indicated by the Brainsight system or Cicerone system.

*Recording techniques.* Multicontact electrodes (V-probes, Plexon, Inc) were lowered using the NAN microdrive system (NAN Instruments) until the target region was reached Following a settling period, all active cells were recorded. This lowering depth was predetermined and calculated with the aid of either Brainsight or Cicerone system to make sure the majority of the contacts on the V-probe were in the gray matter of the recording region. Individual action potentials were isolated on a Ripple Grapevine system (Ripple, Inc.). Neurons were selected for study solely on the basis of the quality of isolation; we never pre-selected based on task-related response properties. Cells were sorted offline with Plexon Offline Sorter (Plexon, Inc.) by hand by MZW and lab technician, Cindy Tu. No automated sorting was used. Neurons were assigned to cOFC*m* vs cOFC*l* prior to any analyses by SRH following PCC connectivity criteria (Fig. 1).

*Statistical analyses: Behavior.* Only trials accompanying the recording sessions were analyzed for the current paper. For choice accuracy, we defined the correct choice as choosing the offer with expected value higher than or equal to that of the alternative offer. Expected value (EV) is the product of stakes multiplied by probability of winning (getting reward, in contrast to getting no reward). Probability of choosing offer 1 as a function of value difference (EV1–EV2) is fitted with generalized linear with logistic transform function and binomial distribution. The error bars indicate 95% confidence intervals from the logistic regression model.

*Statistical analyses: Spectral analyses.* Local field potentials (LFP) were collected during recording sessions along with spike data using the Ripple Grapevine system. LFP data from each contact of the Plexon v-probes were used. Raw data was low-pass filtered at 100 Hz and notch-filtered at 60 Hz. All filtering and frequency-domain (spectral) analyses were conducted in Matlab with Chronux toolbox[90]. Power spectra in all three regions were calculated with all LFP channels. Spike-field coherence was calculated using every combination of each spike train in one area and each channel of LFP in another area. Coherence comparison used non-parametric statistics: Wilcoxon signed rank test and Kruskal–Wallis test, both conducted in Matlab. We used the following bandwidths for analyses: Delta (0.5–5 Hz), Theta (5–10 Hz), Alpha (10–15 Hz), Beta (15–30 Hz), and Gamma (>30 Hz). For coherence comparisons, we calculated the coherence with a frequency-resolved method, such that we re-adjusted the sliding calculation

window widths to be four times the max length for each frequency band. We aligned data to either offer 1 or choice to achieve a better temporal resolution of the coherence tests.

*Statistical analyses: Mutual information.* Mathematically, mutual information is defined as I[X;Y] = H[X]-H[X|Y] = I[Y;X], where *I* is the mutual information between random variables *X* and *Y*. It quantifies the information *X* gives upon observing *Y* and is the same as the information *Y* gives upon observing *X*. Equivalently, it captures how much uncertainty about *X* decreases after learning *Y*, and vice versa. We used the Neuroscience Information Theory Matlab toolbox to calculate the mutual information between two spike trains, one from each brain area of interest[54].

To test whether the mutual information in cOFC*m*-PCC or cOFC*l*-PCC during task was higher than expected chance, we shuffled each single-unit's brain area identity to form shuffled ensembles with the same sizes as the original data. Then we shuffled temporal sequences within ITI and, separately, within active task-time. The temporal shuffling is to test whether the increase in mutual information was above chance level and driven by engaging in the task. We then calculated mutual information based on these shuffled ensembles. We repeated this procedure 500 times and obtained the middle 95% range of the shuffled mutual information as a function of time (Fig. 3F, shaded magenta and cyan for cOFC*m*-PCC and cOFC*l*-PCC circuits, respectively). Thus, any value outside the shaded area is significantly higher/lower than expected by chance.

*Statistical analyses: Encoding.* We used a sliding multiple linear regression to characterize the encoding of all task variables (stakes, probabilities, expected values of offer 1 and offer 2, chosen option, chosen location, whether offer 1 was presented on left vs. right, and choice outcome [win or lose * stakes]). To do so, we took the normalized FR of each neuron, averaged across a 200 ms time bin, and then regressed against task parameters. The sliding procedure slid forward with a 10 ms step size. For offer epochs, we used a multiple linear regression model with stakes, probabilities, and expected values (EV) as predictors. Expected value (EV) is defined as the product of stake and probability. For the rest of the epochs, we used a multiple linear regression model with stakes, probabilities, EV1, EV2, chosen option (offer 1 vs. 2), chosen location (left vs. right), outcome (received outcome, 0 for lost gamble, reward of the stake's size for won gamble), and whether offer 1 appeared on the left or right side of the screen. For later tests looking the expected value tuning for left and right offers, we used a multiple linear regression model with stakes, probabilities, left EV (EV*l*), right EV (EV*r*), chosen option (offer 1 vs. 2), chosen location (left vs. right), outcome (actually received outcome, 0 for lost gamble, reward of the stake's size for won gamble), and whether offer 1 (first appeared offer) appeared on the left or right side of the screen. All predictors were centered and converted to categorical variables when applicable. The response variable, FR, were normalized for each neuron across trials to avoid spurious correlation[91].

Proportion of neurons was calculated based on whether neurons significantly encoded a single parameter of interest. Encoding strength was defined as the t-statistics of each predictor variable from the multiple regression. We used t-statistics since they are not influenced by the actual range of each variable (even though we centered all predictor variables) and are comparable across variables. The comparison of encoding strength across all three regions used the nonparametric Kruskal–Wallis test. Latency was defined as, within the analyzed event window, the time lapsed until the encoding strength of the variable of interest reached the peak for each neuron. Then the peak time for a region was calculated as the median of each neuron's peak time. Latency calculation was based on all neurons and not only the significantly tuned ones. Whether latencies from all three regions were significantly different from one another was tested with generalized linear model (GLM) with a Gamma distribution, due to the fact that timing data, such as latency or reaction time, are better described by a Gamma distribution than a Gaussian distribution.

For mutual inhibition, we took the regression coefficients from the above described multiple regression models for the offer 2 epoch and the choice epoch respectively. Then we correlated the coefficients for offer 1 vs. 2 or EV*l* vs. EV*r* with a Spearman correlation. Spearman correlation is chosen to avoid spurious correlation caused only by a few outliers. The strength of mutual inhibition signal is the Spearman correlation coefficient.

*Statistical analyses: Granger causality.* Granger causality measures how one time series could predict (Granger-cause) another time series, after controlling for the fact that the later time series's early sequences also predicts its own later sequences[92]. Sometimes, calculation of Granger causality is also conditioned on simultaneously observing other potentially confounding time series (Lütkepohl, 2005)[93]. For all Granger causality tests, we first used the Augmented Dickey-Fuller test with the autoregressive model with drift variant (ARD) to determine whether a time series was stationary. Then we used the vector autoregression model to determine the best time lag to use through model comparison (Akaike information criterion) with different time lags. Then the Granger causality test was used on stationary time series or with a correction for non-stationary time series. All significant tests for the Granger causality analysis included and controlled for all possible confounding signals. For example, when testing whether the decidability of

choice option (offer 1 vs. offer 2) in cOFC*m* Granger-caused the decidability of choice location (left vs. right) in PCC, the model also included the decidability of choice option in PCC and cOFC*l*, and, choice location in cOFC*m* and cOFC*l*, as simultaneously observed signals, and thus controlled for their explanatory power to the tested Granger causal relation. In other words, the significant Granger causal relations we reported in the main text and supplementary material, cannot be explained away by other simultaneously measured signals. All Granger causality tests were carried out in Matlab. Matlab functions used: adftest, varm, estimate, summarize, gctest, the Econometrics Toolbox.

*Statistical analyses: Decoding.* We first organized population activity patterns for the training and testing of the linear discriminant analysis (LDA) decoder. For each trial, we aligned the normalized FR of each neuron at the onset of offer 1 presentation and took firing from 500 ms before this onset through 2500 ms after this onset as the offer period (including 500 ms ITI before offer 1, offer 1 epoch, offer 2 epoch, and the first 500 ms of decision-making). We also aligned the normalized FR of each neuron at choice execution (when eye-fixation on the chosen offer passed 200 ms and thus signaled commitment to the choice). Then we took the FR from 1500 ms before this onset through 1500 ms after this onset as the choice period (including 1500 ms pre-choice, outcome delivery, and ITI). We then slid through the offer and the choice periods and generated non-overlapping population activity patterns that were 500 ms in width and tiled the entire offer and choice periods.

Then we followed a fourfold cross validation procedure, which involved training different LDA decoders on 75% of the correct trials to differentiate the chosen option (offer 1 vs. 2), the chosen location (left vs. right), and the expected value of offer 1 (EV1 high vs. low) on each trial. Then we tested the decoder on the other 25% of the correct trials. Decoding accuracy in error trials was obtained by using the same trained LDA decoders to decode all error trials (since none of the error trials were used for training). For EV1 high vs. low, we compared EV1 from each trial to the mean EV of all offers. If the EV1 was larger than or equal to the mean, then it was counted as a high EV1, otherwise low.

*Statistical analyses: Population dynamics.* To measure the dynamics in population neural activities, we first organized our spiking data into population states. We defined the population state as the normalized firing rate of each of all simultaneously recorded neurons, averaged over a 200 ms time bin, in each region. Then we slid across all time points in each trial with a 50 ms step size to calculate population states at each sliding step. We calculated these series of population states for two sets of simultaneously recorded ensembles in cOFC*m*, cOFC*l*, and PCC, one from each subject. We then applied principal component analysis (PCA) to identify a lower-dimensional space to then measure the population dynamics. We first selected and grouped all correct trials based on whether (1) offer 1 or offer 2 was chosen; (2) left or right offer was chosen; and (3) offer 1 was a higher or lower than average value of offers. Then we conducted PCA on the trial averaged population states for each pair of the above-mentioned three pairs of conditions. To make the measures of population dynamics comparable across regions, we defined top-N PC space as the top N PCs that captured at least 70% of the variance. For subject P, N equals 6 in cOFC*m*, 5 in cOFC*l*, and 15 in PCC. For subject S, N equals 3 in cOFC*m*, 5 in cOFC*l*, and 3 in PCC. We then projected trial-averaged or trial-by-trial population states from correct or error trials and each pair of conditions onto this top-N PC space. This projection resulted in pairs of population trajectories corresponding to pairs of conditions in the top-N PC space expanding the whole trial length. We then measured the Euclidean distance at each time point in a trial between the pairs of population trajectories. We used a shuffle procedure in which trials were shuffled across conditions. This shuffle procedure was implemented 1000 times to generate 1000 randomized trial-averaged trajectories for each trial condition, and significance cutoff were set at the top and bottom 2.5% of the shuffled results. For trial-by-trial population state projections that resulted in a pair of two sets of population trajectories (that is, each trajectory corresponded to a specific trial condition), we calculated the adjusted Euclidean distance. The adjusted Euclidean distance is the Euclidean distance across conditions (cross distance) normalized by the Euclidean distance within conditions (self distance / dispersion). Cross distance was defined as the Euclidean distance from one point on one trajectory in one trial condition to all the trial-by-trial trajectories' corresponding time point in the other trial condition. Self distance/dispersion was defined as the Euclidean distance of one point on one trajectory in one trial condition to all the other trial-by-trial trajectories' corresponding time point in the same trial condition.

$$adjusted\ distance = \frac{cross\ distance}{self\ distance} \quad (1)$$

Normalizing the cross distance with self distance controls for the "internal noise level" to make the distance comparable across regions[64]. The distance, or separation, between population trajectories from pairs of trial conditions represents the population neural activity variance devoted to distinguish those trial conditions[14]. Intuitively, it can be interpreted as: the larger the distance/separation between trajectories for different conditions, the more information the variance in this neural population conveys to tell these conditions apart. PCA analysis and Euclidean distance calculation used pca and pdist2 functions in Matlab.

**Reporting summary**. Further information on research design is available in the Nature Research Reporting Summary linked to this article.

## Data availability

The data are available under restricted access for interpreting and verifying the research in this article. The behavioral and sorted neural data generated in this study have been deposited in the Open Science Framework (OSF) database under https://osf.io/npqhg/?view_only=7ce471770ed54e44b3caf0a7f5d0132d.

## Code availability

All code is available at the request of the reader from the corresponding author.

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

## Acknowledgements

We thank Giuliana Loconte, Hannah Lee, Tanya Casta, Cindy Tu, Mark Grier, Megan Monko, and Adriana Cushnie for experimental help. This research was supported by NIH grants R01 DA038106 (to B.Y.H.), R01 MH 118257 (to S.R.H.)., and a MNDrive fellowship (to M.Z.W.).

## Author contributions

M.Z.W. performed the analyses of the data; M.Z.W., B.H., and S.H. contributed to the development of the hypotheses and the writing of the text.

## Competing interests

The authors declare no competing interests.
