## [Peer Review File · Nature Communications]

A structural and functional subdivision in central orbitofrontal cortexREVIEWER COMMENTS

Reviewer #1 (Remarks to the Author):

In the current study, Wang et al. investigated the functional connectivity between subregions of the central OFC and PCC in a value-based decision task. The central OFC is often considered as a whole in many studies, but in reality, it may be further divided into many functional distinct subregions. This study may help us to understand the functional differences between these subregions.

Major:

1. My biggest concern is the small number of neurons that were recorded. 44 cells in cOFCm (23 from subject P, 21 from subject S), 54 cells in 138 cOFCI (28 from subject P, 26 from subject S) and 213 cells in PCC (89 from subject P, 124 from 139 subject S). OFC is a big area, OFC neurons are heterogeneous, and their firing rates are low. These numbers are hard to justify many small effects in the later analyses. The authors in general need to provide better statistics for their analyses. For example, the authors need to show the effects are not caused by one or two “good” neurons. Bootstrap/resampling may be used to test the robustness of these findings.
2. The tracing suggests a difference in the projection pattern between PCC gyrus and sulcus. It would be nice if the authors could show a related difference in the electrophysiology.
3. Regarding analyses in Fig 3A-F, is the negative correlation of value encoding between offer 1 and offer 2 in the offer 2 epoch due to a reversed encoding of offer 1 or offer 2 during the offer 2 period when compared to the encoding of offer 1 during the offer 1 period?

Minor:

1. The introduction doesn't provide a strong enough rationale for studying the functional connectivity between the two OFC subregions and the PCC. For example, why PCC instead of LPFC?
2. Fig 1D: it would be helpful to use separate colors for cOFCm and cOFCI. How did the authors determine the area for neurons that sit near the border between cOFCm and cOFCI.
3. Fig 4 A-F these plots are too messy to see anything useful. Too many colors. Maybe use solid and dash lines to indicate choices 1 & 2 separately? Also, in the legend: '+' individual outliers. I don't see any + in the plots.
4. The last paragraph on page 19 is hard to decipher: “Our results point to one possible case of this distinction, where some OFC value signals are relatively abstract and others are relatively concrete, but the two value representations are mutually reinforcing. More speculatively, our results suggest that even apparent intra-areal redundancy of function may mask an underlying heterogeneity of function.” What exactly are the abstract and concrete signals? What do authors mean by saying “mutually reinforcing”?
5. Pg 10, line 211-212: “than chosen option” is repeated twice.

Reviewer #2 (Remarks to the Author):

This manuscript reports on a single-unit recording experiment examining how economic choice is processed in the central orbitofrontal (cOFC) and posterior cingulate cortices (pCC) of macaque monkeys. The authors performed an extensive series of sophisticated analyses testing the hypothesis that the medial central OFC compares competing offers in value space and sends this information to pCC, where it is transformed into action space, a transformation that is in theory required to ultimately execute a decision. A very nice control in this study is that the authors also recorded in a more lateral part of medial OFC and found that this region did not show the same relationship with pCC. Also reported in the manuscript is a tract-tracing experiment testing the bidirectional connections between cOFC and pCC. Both the hypothesis and the analyses are compelling, and overall the paper is excellent. I have a few comments and questions.

1. The authors are somewhat vague about exactly where the recording sites in cOFC, medial section and lateral sections, were located. Figure 1D is said to show “example recording sites” rather than depicting the extent of cortex from which recordings characterized as medial and lateral were located.

Although it is clear that by medial cOFC, the authors mean the cortex within the medial orbital sulcus, they never specify what exactly they mean by lateral cOFC. It is important to be as specific as possible about where the recordings were located and where lateral cOFC ends, especially because these terms seem to represent new anatomical subdivisions.

2. The negative correlation of regression coefficients shown in Figure 3 are very interesting. However, there is an ambiguity in what EV1 and EV2 mean (e.g. in Figure 3A), and I have trouble resolving this ambiguity by reading the Methods. Do EV1 and EV2 mean the greater and lesser expected value offers, or the first and second offers? The authors suggest that the cOFC(medial) represents the offers in “value space” which implies that the negative correlation should be between the regression coefficients of firing rate with the larger value offer (EV1) against the regression coefficients of firing rate with the lower value option (EV2). But I can’t tell if this is what is meant here. If it is instead the first and second offer values, that wouldn’t exactly be “value space” but something more like “sequence space,” by analogy with what the pCC is said to represent, which is “action space”, because the regression coefficients against the left and right offers are what are negatively correlated – and the left and right offers require different actions to be chosen. This ambiguity should be clarified in the paper. If it is the first and second offers that are meant by EV1 and EV2, the authors should justify how this relates to value space.

3. One obvious kind of analysis that was not done was to test how the distance of the EV comparison (i.e. large EV differences between the two offers vs small EV differences) related to the correlation of regression coefficients and the separation of population trajectories. The authors’ hypothesis predicts that these neural effects should be harder to achieve, or more important, when the offer values are close, and easier to achieve, and less important, when the offer values are farther apart. The apparent transfer of information from cOFC to pCC would presumably be different according to the difference in EV value between the offers. That is, rather than categorically comparing errors with correct choices, as the authors did, they could have done a fuller analysis across the parametric space of EV difference (some “correct” choices are more correct than others). I am not arguing that this should be a required analysis for publishing this paper, but I think it would have been obvious and informative to look at this.

4. One somewhat theoretical issue that the authors seem to evade has to do with the back propagation of information from pCC to cOFC that they report. It is fascinating that this back propagation occurs later than what might be called the “forward” propagation from cOFC to pCC, suggesting some kind of feedback. But for feedback to OFC to be useful, it would need to be based on the outcome of the choice. If the representation of the prospective choice in “action space” in pCC were simply strengthening the representation of the choice in “value space”, where it came from in the first place, it would be pointless and perhaps counterproductive, simply reinforcing itself internally. Can the authors shed a little more light about when this back propagation of information occurs and how they are interpreting it functionally? Do they think it represents meaningful feedback, or could it just be some kind of artifact? For example, does strong feedback help later choices to become more accurate?

Minor points

5. Many of the analyses are not well described in the results. While it is expected to reserve full analytical details for the Methods section, it would help the readability of the paper to provide a short intuitive summary of each analysis being done, along with its rationale, in the results. For example, the authors provide no explanation in the results for why a negative correlation between regression coefficients for the two offers might be a “signature of comparison,” but instead only refer to their previous papers. Surely, there is one-sentence description that would convey this idea for those who do not remember the authors’ other work. Another example has to do with the Granger analyses. A short summary of what this means and an intuitive idea of how it is done would be helpful to general readers. This could only help the paper.

6. How does the coherence analysis relate to errors vs correct choices? It is not obvious whether this analysis was done using all trials, or only correct ones.

7. On page 17, there are two sudden references to how oscillatory activity bands relate to the transfer

of information between cOFCm and pCC and vice versa. For example, on line 347-8, "coinciding with theta oscillations," and on line 350, "coinciding with delta oscillations." It is not clear where these ideas come from – are they derived from data in this paper, or are they speculations based on the literature? If they are the latter, then they would be more appropriate for the discussion section, whereas if they are the former, what data they are based on should be made clearer.

Reply to reviewers

Reviewer #1:

In the current study, Wang et al. investigated the functional connectivity between subregions of the central OFC and PCC in a value-based decision task. The central OFC is often considered as a whole in many studies, but in reality, it may be further divided into many functional distinct subregions. This study may help us to understand the functional differences between these subregions.

We appreciate that the reviewer understands the importance of these functionally and anatomically distinct OFC regions.

Major:

1. My biggest concern is the small number of neurons that were recorded. 44 cells in cOFCm (23 from subject P, 21 from subject S), 54 cells in 138 cOFCI (28 from subject P, 26 from subject S) and 213 cells in PCC (89 from subject P, 124 from 139 subject S). OFC is a big area, OFC neurons are heterogeneous, and their firing rates are low. These numbers are hard to justify many small effects in the later analyses. The authors in general need to provide better statistics for their analyses. For example, the authors need to show the effects are not caused by one or two “good” neurons. Bootstrap/resampling may be used to test the robustness of these findings.

We understand the reviewer’s concern. We appreciate that the reviewer provides helpful suggestions for remedying this concern. We have addressed it in three ways, including the one suggested by the reviewer, and all of which have now been added to the Supplementary Material: (1) Monte-Carlo resampling, (2) power analysis, and (3) outlier removal. All of these point to the robustness of our results. Here is the added text:

Robustness of findings

We verified that we had sufficient numbers of neurons in each area to perform these analyses in the following three ways:

First, we performed an outlier analysis. In the original analyses, we used Spearman correlation to examine the relationship between regression coefficients for EV1 vs. EV2 and for EVI vs. EVr, because this analysis is insensitive to outliers. This approach is more robust to outliers than the more common Pearson correlation. To further confirm that our results were not driven by any detectable outliers, we first used Cook’s D to measure the global influence (both discrepancy and leverage) for each pair of regression coefficient sets in the correlation analyses in Figure 3A-F. By this method, we detected a single cell as an outlier in the analysis presented in Figure 3A. Specifically, with the outlier (as in Fig 3A), the Spearman correlation coefficient ρ is -0.36 ($p=0.016$). After removing the outlier and repeating the same analysis, we found

rho=-0.319 (p=0.038), which is also statistically significant. These two correlation coefficients are not significantly different from each other, indicating that the presence of an outlier did not itself have a measurable significant effect (z=0.208, p=0.835, Fisher's transformation test). For the analysis depicted in Figure 3C, we detected another cell as an outlier. With the outlier, the Spearman correlation coefficient rho is 0.02 (p=0.943). After removing the outlier and repeating the same analysis, we found rho=-0.019 (p=0.782). These two correlation coefficients are not significantly different (z=-0.401, p=0.689, Fisher's transformation test). These results indicate that our null finding was not driven by outliers.

Second, we used Monte-Carlo resampling to generate a 200-neuron pseudo-ensemble by randomly resampling the cOFC_m dataset (with replacement). Then, we repeated the analyses described in the main text on the cOFC_m pseudo-ensemble. In particular, we obtained Spearman correlation coefficients between regression weights of firing rates against the EV1 and EV2 parameters. We repeated this resampling and reanalysis 1000 times to obtain a distribution of these resampled correlation coefficients (the red distribution in Supplementary Figure 5). We take the mean of this distribution as an estimate of the true, underlying Spearman correlation coefficient. Notably, the observed Spearman correlation using the original non-resampled data was near the center of this bootstrapped distribution, and, indeed, was not significantly different from its mean. These results indicate that, despite its small sample size, our observed data were close to the distribution we would expect with a larger dataset. We then repeated the same procedure for cOFC_l (the orange distribution in Supplementary Figure 5). As expected, we found that the true Spearman was also not significantly different from the bootstrapped re-estimate. We repeated the same analysis for EVI vs EV_r in cOFC_m and cOFC_l. We found that the true cOFC_m data were not significantly different from the bootstrapped re-estimate. The Spearman correlation coefficient from recorded cOFC_l data was slightly higher than the population estimation. However, this does not change our original finding: the distribution of the population estimation for cOFC_m is still significantly different from that for cOFC_l (KS stat = 0.76, p<0.001, Kolmogorov-Smirnov test). Crucially, this result confirms that the Spearman correlation coefficients for encoding formats of EVI and EV_r form two significantly different distributions in cOFC_m and cOFC_l, suggesting that neurons from these two cOFC subregions perform significantly different neural computations for representing EVI and EV_r.

Supplementary Figure 5: Histograms showing range of values of resampled data and their overlap. A. Red distribution: 1000 resamples (with repeats) of EV1 vs EV2 distribution for cOFC_m. Orange distribution: same but for cOFC_l. The cOFC_l distribution has a significantly higher correlation than the cOFC_m. B. Same as panel A, but with the EVI vs. EV_r variables. Dark blue color: cOFC_m; Light blue: cOFC_l.

Third, we performed a power analysis. To estimate the effect size, we used the mean effect size of a previous study (Wang et al., 2017) from our lab that recorded in cOFC and conducted the same ensemble analysis as in the current study. In this previous study, the median effect size of significant correlations between two sets of regression coefficients was $r = 0.33$ (effect sizes of the significant correlations reported in the paper: 0.68, 0.33, 0.41, 0.31, 0.27, 0.36 and 0.2). We used 0.05 as significance level and 0.60 as power. For cOFC_m, a power analysis with these parameters suggests that the minimum sample size required to detect an effect size of -0.36 (in Figure 3A) with significance level 0.05 and power 0.60 is $n = 44$. Relatedly, with a sample size of 44 neurons in cOFC_m, significance level of 0.05, and power of 0.60, the effect size we are expecting is 0.329. Similarly, for cOFC_l, a sample size of 54, significance level of 0.05, and power of 0.60, the effect size we are expecting is 0.298. These results indicate that our study was sufficiently powered to detect the effects that we report.

2. The tracing suggests a difference in the projection pattern between PCC gyrus and sulcus. It would be nice if the authors could show a related difference in the electrophysiology.

Based on this suggestion, we performed a new control analysis in which we separated recorded PCC neurons into PCC gyrus (PCCg) and PCC sulcus (PCCs) based on the placement of each recording contact. Then we repeated the ensemble analysis shown in Figures 3C and 3F on PCCg and PCCs separately, instead of on the combined PCC ensemble. The following text has been added to the Supplementary Material:

No differences between PCC gyrus and sulcus

Based on this connectivity differences observed between M1FR and M6FR, we separated recorded PCC neurons into PCC gyrus (PCCg) and PCC sulcus (PCCs) based on the placement of each recording contact. Then we conducted the ensemble analysis shown in Figures 3C and 3F on PCCg and PCCs separately, instead of on the combined PCC ensemble.

The correlation coefficient between encoding formats for EV1 and EV2 is -0.024 ($p = 0.847$, Spearman correlation) in PCCg, and 0.043 ($p = 0.526$) in PCCs. These two correlation coefficients are not significantly different from each other ($z = -0.473$, $p = 0.637$, Fisher's Transformation test). Moreover, neither the coefficient in PCCg ($z = -0.320$, $p = 0.749$) nor that in PCCs ($z = 0.253$, $p = 0.800$) was significantly different from the coefficient in the combined PCC ensemble (in Figure 3C).

The correlation coefficient between encoding formats for EVl and EVr is -0.400 ($p < 0.001$, Spearman correlation) in PCCg, and -0.151 ($p = 0.024$) in PCCs. These two correlation coefficients are not significantly different from each other ($z = -1.934$, $p = 0.053$, Fisher's Transformation test). Moreover, neither coefficient in PCCg ($z = -1.314$, $p = 0.189$) nor that in PCCs ($z = 1.037$, $p = 0.300$) was significantly different from the coefficient in the combined PCC ensemble (in Figure 3C).

Although we do expect functional differences between PCCs and PCCg, perhaps they would not be reflected in the particular framework studied here. In addition, the connectivity-based division identified, based on injection sites, may be too coarse. Future injections in the OFC will hopefully help to clarify the true divisions.

3. Regarding analyses in Fig 3A-F, is the negative correlation of value encoding between offer 1 and offer 2 in the offer 2 epoch due to a reversed encoding of offer 1 or offer 2 during the offer 2 period when compared to the encoding of offer 1 during the offer 1 period?

The reviewer has asked an interesting question. That question was not answered in the original text. To answer it, we have performed the appropriate analyses and added the following paragraph to the Results:

We next asked whether the mutual inhibition signal between offer 1 and offer 2 (significantly observed in cOFC_m) during the offer 2 epoch reflects a reversal of encoding format for offer 1 from the offer 1 epoch to the offer 2 epoch. We found that in cOFC_m, offer 1 in the offer 1 epoch was encoded in the same format as offer 1 in the offer 2 epoch ($r = 0.385$, $p = 0.010$). Consistent with the reported mutual inhibition signal in Fig 3A, the encoding format of offer 1 during the offer 1 epoch was also negatively correlated with encoding format of offer 2 during the offer 2 epoch ($r = 0.314$, $p = 0.039$). These results suggest that offer 1 encoding format did not change in cOFC_m from the offer 1 to the offer 2 epochs. Rather, offer 2 used a reverse format to that of offer 1 when it came online in the offer 2 epoch.

We further examined the offer 1 encoding format in cOFC_l and PCC. In cOFC_l, offer 1 in the offer 1 epoch was encoded in a format uncorrelated with offer 1 in the offer 2 epoch ($r = 0.078$, $p = 0.578$). Consistent with the reporting in Fig 3B, encoding format of offer 1 during the offer 1 epoch was uncorrelated with encoding format of offer 2 during the offer 2 epoch ($r = 0.224$, $p = 0.104$). In PCC, offer 1 in the offer 1 epoch was encoded in the same format as offer 1 in the offer 2 epoch ($r = 0.271$, $p < 0.001$). Consistent with data shown in Fig 3C, encoding format of offer 1 during the offer 1 epoch was uncorrelated with encoding format of offer 2 during the offer 2 epoch ($r = 0.065$, $p = 0.344$).

Minor:

1. The introduction doesn't provide a strong enough rationale for studying the functional connectivity between the two OFC subregions and the PCC. For example, why PCC instead of LPFC?

This is a fascinating question, and we appreciate having the opportunity to expand on it. We chose these because the PCC because of our interest in spatial information and how it affects choice – there is plentiful evidence linking PCC to space and value, and our past work was centered on that theory. This work extends those ideas. PCC is in a position to facilitate integration of choice related information from cOFC with spatial information from the parietal and medial temporal lobes. We have clarified this in the Introduction. However, we think that the reviewer is also highlighting a deeper point: there are many other pairs of connected/unconnected regions that could have been studied and that ought to be studied in the future. We have added text to the Discussion hypothesizing these possible functional relationships.

2. Fig 1D: it would be helpful to use separate colors for cOFC_m and cOFC_l.

We have re-worked Figure 1 to more easily convey connectivity, recording sites, and putative divisions. We use the colors green and orange to indicate cOFCI and cOFCm respectively. We hope that the new version is easier to digest. We have reprinted it here:

How did the authors determine the area for neurons that sit near the border between cOFCm and cOFCI.

Determining the area is done by classifying neurons according to their specific position relative to the anatomically defined borders. Our methods, which use the Brainsight technique, give an error of less than 1 mm in the X and Y dimensions. The major concern with this kind of question, of course, is whether the borders were drawn post-hoc, to give the desired answer, would could produce false positive results. It is critical to note, then, that all neurons were assigned to cOFCm vs cOFCI prior to performing any electrophysiological analyses, based on the divisions above. This fact is now noted in the revised Methods.

3. Fig 4 A-F these plots are too messy to see anything useful. Too many colors. Maybe use solid and dash lines to indicate choices 1 & 2 separately? Also, in the legend: '+' individual outliers. I don't see any + in the plots.

Excellent point. We have re-worked Figure 4 (pasted below).

4. The last paragraph on page 19 is hard to decipher: “Our results point to one possible case of this distinction, where some OFC value signals are relatively abstract and others are relatively concrete, but the two value representations are mutually reinforcing. More speculatively, our results suggest that even apparent intra-areal redundancy of function may mask an underlying heterogeneity of function.” What exactly are the abstract and concrete signals? What do authors mean by saying “mutually reinforcing”?

We apologize for the unclear writing. On revision, given other comments, we have removed this paragraph (and also expanded on others in the Discussion).

5. Pg 10, line 211-212: “than chosen option” is repeated twice.

Fixed.

Reviewer #2:

This manuscript reports on a single-unit recording experiment examining how economic choice is processed in the central orbitofrontal (cOFC) and posterior cingulate cortices (pCC) of macaque monkeys. The authors performed an extensive series of sophisticated analyses testing the hypothesis that the medial central OFC compares competing offers in value space and sends this information to pCC, where it is transformed into action space, a transformation that is in theory required to ultimately execute a decision. A very nice control in this study is that the authors also recorded in a more lateral part of medial OFC and found that this region did not show the same relationship with pCC. Also reported in the manuscript is a tract-tracing experiment testing the bidirectional connections between cOFC and pCC. Both the hypothesis and the analyses are compelling, and overall the paper is excellent. I have a few comments and questions.

We appreciate these kind words about our study.

1. The authors are somewhat vague about exactly where the recording sites in cOFC, medial section and lateral sections, were located. Figure 1D is said to show “example recording sites” rather than depicting the extent of cortex from which recordings characterized as medial and lateral were located. Although it is clear that by medial cOFC, the authors mean the cortex within the medial orbital sulcus, they never specify what exactly they mean by lateral cOFC. It is important to be as specific as possible about where the recordings were located and where lateral cOFC ends, especially because these terms seem to represent new anatomical subdivisions.

We have addressed this in two ways. First, we have revised Figure 1 substantially to make clear the extent of the different regions, based on anatomical connectivity (see also the response to R1). Second, we have clarified the extent of the recording sites within these regions in the text. The revised figure is shown here:

2. The negative correlation of regression coefficients shown in Figure 3 are very interesting. However, there is an ambiguity in what EV1 and EV2 mean (e.g. in Figure 3A), and I have trouble resolving this ambiguity by reading the Methods. Do EV1 and EV2 mean the greater and lesser expected value offers, or the first and second offers?

We apologize for the lack of clarity. EV1 refers to the expected value of the first offer presented in the series, and EV2 refers to the expected value of the second offer presented in the series. Because the order of offers was randomized, EV1 was greater only half the time; the rest of the time, EV2 was greater. We have now revised the text for clarity at multiple locations.

The authors suggest that the cOFC(medial) represents the offers in “value space” which implies that the negative correlation should be between the regression coefficients of firing rate with the larger value offer (EV1) against the regression coefficients of firing rate with the lower value option (EV2). But I can’t tell if this is what is meant here. If it is instead the first and second offer values, that wouldn’t exactly be “value space” but something more like “sequence space,” by analogy with what the pCC is said to represent, which is “action space”, because the regression coefficients against the left and right offers are what are negatively correlated – and the left and right offers require different actions to be chosen. This ambiguity should be clarified in the paper. If it is the first and second offers that are meant by EV1 and EV2, the authors should justify how this relates to value space.

The reviewer's point is valid and we apologize for the confusion. We have revised the text throughout to accommodate this change. Specifically, following the reviewer, we use the term “abstract sequence space”. We note that the distinction between "abstract value based" and "order based" is not critical to our arguments - what is important is that the OFC representation is non-spatial and that the PCC one is spatial. This does not change.

3. One obvious kind of analysis that was not done was to test how the distance of the EV comparison (i.e. large EV differences between the two offers vs small EV differences) related to the correlation of regression coefficients and the separation of population trajectories. The authors’ hypothesis predicts that these neural effects should be harder to achieve, or more important, when the offer values are close, and easier to achieve, and less important, when the offer values are farther apart. The apparent transfer of information from cOFC to pCC would presumably be different according to the difference in EV value between the offers. That is, rather than categorically comparing errors with correct choices, as the authors did, they could have done a fuller analysis across the parametric space of EV difference (some “correct” choices are more correct than others). I am not arguing that this should be a required analysis for publishing this paper, but I think it would have been obvious and informative to look at this.

We fully agree with and thank the reviewer for this suggestion. To address this comment, we separated all correct trials into easy and difficult choice conditions. The easy condition contains trials in which the EVs of the two options are far apart; the difficult choices are those for which the EVs of the two options are

close together. The median of the absolute values of the differences between the two offers served as the dividing line. (At the same time, we also acknowledge and agree with the reviewer that the 'difficult' choices are not necessarily difficult, since when two values are close, choosing either could be good enough.)

The value comparison signal between EV1 and EV2 (correlation coefficient between regression coefficients for EV1 vs. EV2) in cOFCm Granger-caused the value comparison signal between EVl and EVr in PCC in both easy choice conditions ($gc=100.75$, $p = 0.017$) and difficult choice conditions ($gc=116.51$, $p = 0.016$). However, the Granger causal relation emerged 140 ms earlier in easy relative to difficult choice conditions. This result suggests that easy choices potentially take less time to compare and thus lead to faster transfer of choice information from cOFCm in a more abstract framework to PCC to a more concrete, action-based framework.

This new finding has been added to the Results and Supplementary Material.

4. One somewhat theoretical issue that the authors seem to evade has to do with the back propagation of information from pCC to cOFC that they report. It is fascinating that this back propagation occurs later than what might be called the "forward" propagation from cOFC to pCC, suggesting some kind of feedback. But for feedback to OFC to be useful, it would need to be based on the outcome of the choice. If the representation of the prospective choice in "action space" in pCC were simply strengthening the representation of the choice in "value space", where it came from in the first place, it would be pointless and perhaps counterproductive, simply reinforcing itself internally. Can the authors shed a little more light about when this back propagation of information occurs and how they are interpreting it functionally? Do they think it represents meaningful feedback, or could it just be some kind of artifact? For example, does strong feedback help later choices to become more accurate?

The reviewer asks some fascinating questions that, unfortunately, go beyond our ability to make definitive statements. Having said that, we think it is reasonable to make educated guesses about the meaning of this signal, as long as they are carefully couched as speculation. In short, we agree with the reviewer's guess, even as we acknowledge that it is just one possible interpretation. We have added the following new text to the manuscript to do so:

Our data suggest that there may be important information traveling from the PCC to the cOFC, and that this information transfer may occur after the transfer of information from cOFC to PCC. It is reasonable, then, to wonder what function this back-transfer serves. Absent causal manipulations, it is impossible to offer a definitive answer. However, our data do provide enough information for us to make an educated speculation. Specifically, we conjecture that the transmission of information from PCC to cOFC can facilitate the process of reaching a consensus within OFC. We have proposed in the past that decisions in core economic regions can occur gradually, and that it is possible to detect partially completed decisions (Azab and Hayden, 2017, 2020). These partially completed decisions could then be transmitted to other regions and, in turn, influence the ongoing decision. While this idea is consistent with our data, our data nonetheless do not offer strong evidence in its favor; as such, testing this hypothesis remains an important future goal.

Minor points

5. Many of the analyses are not well described in the results. While it is expected to reserve full analytical details for the Methods section, it would help the readability of the paper to provide a short intuitive summary of each analysis being done, along with its rationale, in the results. For example, the authors provide no explanation in the results for why a negative correlation between regression coefficients for the two offers might be a “signature of comparison,” but instead only refer to their previous papers. Surely, there is one-sentence description that would convey this idea for those who do not remember the authors’ other work. Another example has to do with the Granger analyses. A short summary of what this means and an intuitive idea of how it is done would be helpful to general readers. This could only help the paper.

We have added these explanations in the appropriate places in the text, at several points. For the two specific ones mentioned by the reviewer, these are as follows:

The reason this is a putative signal of value comparison is that it reflects a coding of the difference in the values of the two options – the key decision variable for choice, because it can be rectified to produce choice (Hayden and Moreno-Bote, 2018).

We next used Granger causality (see Methods), a method that examines the relative correlation between two time series at different lags to identify putative a causal role between the two, given certain assumptions.

6. How does the coherence analysis relate to errors vs correct choices? It is not obvious whether this analysis was done using all trials, or only correct ones.

We only used correct trials in the coherence analysis. (We now state this clearly in the revised text).

Incidentally, we were interested in the functional connectivity between OFC and PCC when they function together to solve the choice problem. We excluded the error trials since it is not clear whether the error is correlated with changes in the coherence between OFC and PCC. In addition, we do not have matched numbers of error and correct trials, since monkeys showed higher choice accuracy and rarely made mistakes. Moreover, the reason for error is hard to estimate -- break of attention, difficulty in choice, etc. These reasons make it harder to examine error trials alone.

7. On page 17, there are two sudden references to how oscillatory activity bands relate to the transfer of information between cOFCm and pCC and vice versa. For example, on line 347-8, “coinciding with theta oscillations,” and on line 350, “coinciding with delta oscillations.” It is not clear where these ideas come from – are they derived from data in this paper, or are they speculations based on the literature? If they are the latter, then they would be more appropriate

for the discussion section, whereas if they are the former, what data they are based on should be made clearer.

We agree with the reviewer and have moved this point to the Discussion, since it is a possible interpretation of the data shown.

REVIEWER COMMENTS

Reviewer #1 (Remarks to the Author):

I appreciate the authors' detailed responses to my comments. The authors have adequately addressed my concerns, and I have no further comments.

Reviewer #2 (Remarks to the Author):

This manuscript represents a revision of one I previously reviewed. The authors have done a pretty good job in responding to my comments and revising the manuscript, which is improved, but I have several remaining comments:

1. The new paragraph from line 243 to 258 is extremely confusing and poorly written. It would help to introduce this paragraph with a better description of the overall idea. The reader gets lost in a sea of "offer 1 format in offer 2 and offer 2 format in offer 1" (I'm paraphrasing), so that he or she loses track of the overall point. And beyond a new introductory sentence, it would also help to go over the text carefully and try to be clearer about what each sentence means.
2. In the paragraph about neural computation (starting on line 214), I'm not clear on what the latencies are measured from. So the cOFCm encodes the chosen option with a latency of 90ms – but it is not stated what that latency is from.
3. Later in that same paragraph, it is stated that PCC encodes chosen location in more neurons and with a shorter latency than either cOFC region (lines 222-226). The values for the percentages of neurons in each cOFC region and latency in PCC vs each OFC region are not stated (the Supplementary material is referenced). This information is relevant and would take up very little space (four percentages and three latencies) and the reader should not have to dig for it in the Supplemental. Why not just state it there?
4. In the Granger analyses described starting on page 13 (starting on lines 278), it is not clear which epochs the data came from. Earlier in the text when describing a different analysis, the authors wrote "We performed this analysis using a 200-ms analysis window (350 ms after offer 2 onset; the same window identified by the Granger analysis, see below)." For one, this information should also be stated when the Granger analysis is itself introduced. And secondly, it is not clear what is meant by saying that this window was identified by the Granger analysis. The Granger analysis identifies time-lags, not necessarily particular epochs, does it? I would think you would have to choose which epoch you use for the Granger analysis – can you please clarify which epoch you chose for this and why? This gets at the question from my first review as to when the back propagation of information from PCC to cOFCm occurs. I appreciated the authors answer to the question of what the function of this back propagation might be – that was interesting – but I still don't understand what the timing of this back propagation is. Knowing which epoch the Granger analysis was performed on (without digging in the Methods) would help clarify this.

Reply to reviewers

Reviewer #2 (Remarks to the Author):

1. The new paragraph from line 243 to 258 is extremely confusing and poorly written. It would help to introduce this paragraph with a better description of the overall idea. The reader gets lost in a sea of “offer 1 format in offer 2 and offer 2 format in offer 1” (I’m paraphrasing), so that he or she loses track of the overall point. And beyond a new introductory sentence, it would also help to go over the text carefully and try to be clearer about what each sentence means.

We apologize for the poor quality of the writing. The following text, written with clarity in mind, replaces the problematic paragraph:

This negative correlation between regression weights, then, is a putative neural correlate of value comparison through mutual inhibition. We wondered whether the transition of attention from offer 1 to offer 2 results in a reversal of tuning for offer 1 value, as predicted by attentional alignment models of value encoding (Hayden and Moreno-Bote, 2018; McGinty et al., 2016; Krajbich et al., 2010). We found that in cOFC_m, the relevant betas are positive correlated ($r = 0.385$, $p = 0.010$). Likewise, the regression weights for offer 1 during the offer 1 epoch were also negatively correlated with those for offer 2 during the offer 2 epoch ($r = -0.314$, $p = 0.039$). Consistent with the idea that the two cOFC subregions are functionally different, the pattern was different in cOFC_l – specifically, no correlation was observed for either comparison (respectively: $r = 0.078$, $p = 0.578$) and $r = 0.224$, $p = 0.104$). The corresponding data in PCC resembled the patterns in cOFC_m for the first comparison, although not for the second ($r = 0.271$, $p < 0.001$; $r = 0.065$, $p = 0.344$). Overall, these results highlight the differences between cOFC_l and cOFC_m, specifically, that the putative neural correlate of value comparison is observed in the medial area, but not detected in the lateral area.

2. In the paragraph about neural computation (starting on line 214), I’m not clear on what the latencies are measured from. So the cOFC_m encodes the chosen option with a latency of 90ms – but it is not stated what that latency is from.

We apologize that this information was not given. The latency is defined in terms of delay relative to the appearance of offer 2. We now note this information in the revised text.

3. Later in that same paragraph, it is stated that PCC encodes chosen location in more neurons and with a shorter latency than either cOFC region (lines 222-226). The values for the percentages of neurons in each cOFC region and latency in PCC vs each OFC region are not stated (the Supplementary material is referenced). This information is relevant and would take up very little space (four percentages and three latencies) and the reader should not have to dig for it in the Supplemental. Why not just state it there?

We have added this information. Specifically:

Neither OFC region shows this pattern (cOFCm: 18.8% encoding location, 18.8% encoding option; cOFC/: 13.0% encoding location, 16.7% encoding option, see Supplementary material).

In addition, PCC (140 ms) and cOFCm (150 ms) encoded the chosen location with significantly shorter latencies than cOFC/ (230 ms; $F=5.71$, $p=0.004$; Supplementary material).

4. In the Granger analyses described starting on page 13 (starting on lines 278), it is not clear which epochs the data came from. Earlier in the text when describing a different analysis, the authors wrote “We performed this analysis using a 200-ms analysis window (350 ms after offer 2 onset; the same window identified by the Granger analysis, see below).” For one, this information should also be stated when the Granger analysis is itself introduced.

We now do so.

And secondly, it is not clear what is meant by saying that this window was identified by the Granger analysis. The Granger analysis identifies time-lags, not necessarily particular epochs, does it? I would think you would have to choose which epoch you use for the Granger analysis – can you please clarify which epoch you chose for this and why? This gets at the question from my first review as to when the back propagation of information from PCC to cOFCm occurs. I appreciated the authors answer to the question of what the function of this back propagation might be – that was interesting – but I still don't understand what the timing of this back propagation is. Knowing which epoch the Granger analysis was performed on (without digging in the Methods) would help clarify this.

The analysis epoch here was the entire period of interest – beginning with the start of the second offer epoch and ending at the time at which the choice was overtly indicated through the initiation of a saccade.

We now state this here directly: “For the period of analysis, we used the whole period of interest, that is, an epoch beginning with the appearance of the second offer and ending with the occurrence of the choice, as indicated by the start of a saccade towards the choice target”.

The confusion here is our fault – our phrasing was ambiguous. We have removed the text saying the window was identified by the Granger analysis – that does not describe what we did. Instead, the reviewer's guess – that we predefined our period of analysis – is precisely what we did. We now say that.

REVIEWER COMMENTS

Reviewer #2 (Remarks to the Author):

Thank you to the authors for responding to my few remaining comments. These have been addressed fully. I would just point out that on page 13, the two new sentences specifying the time epoch used for the Granger analysis (lines 277-280) seem to contradict each other. It might be wise to resolve or clarify this apparent contradiction.